# Effects of Adding *Sphingomonas* Z392 to Drinking Water on Growth Performance, Intestinal Histological Structure, and Microbial Community of Broiler Chickens

**DOI:** 10.3390/ani14131920

**Published:** 2024-06-28

**Authors:** Mingcheng Wang, Jie Zhong, Yanan Guo, Shuqiang Zhao, Huili Xia, Gailing Wang, Chaoying Liu, Aizhen Guo

**Affiliations:** 1National Laboratory of Agricultural Microbiology, Wuhan 430070, China; wangmingcheng@huanghuai.edu.cn; 2College of Veterinary Medicine, Wuhan 430070, China; 3Hubei Hongshan Laboratory, Huazhong Agricultural University, Wuhan 430070, China; 4College of Biological and Food Engineering, Huanghuai University, Zhumadian 463000, China; zhongjie@huanghuai.edu.cn (J.Z.); hhxyxhl@163.com (H.X.); wanggailing@huanghuai.edu.cn (G.W.); liuchaoying@huanghuai.edu.cn (C.L.); 5Animal Science Institute, Ningxia Academy of Agriculture and Forestry Sciences, Yinchuan 750002, China; gyn330@126.com; 6Animal Disease Prevention and Quarantine Center of Zhumadian City, Zhumadian 463000, China; zq888777@126.com

**Keywords:** broiler, *Sphingomonas*, EPI, intestinal histology, intestinal microbiota

## Abstract

**Simple Summary:**

After adding 4 × 10^5^ CFU/mL of *Sphingomonas* sp. Z392 to drinking water, through the improvement of intestinal histology and flora structure, the digestive and absorption functions of broiler chickens was enhanced, thereby improving their feed utilization and growth performance. Consequently, the final weight of the broilers increased by 4.33%, and the EPI increased by 10.10%, which led to there being more economic benefits from the broilers. Combined with the ammonia-reducing function of this strain, the use of *Sphingomonas* sp. Z392 in broiler breeding is expected to provide a new strategy for the sustainable development of the broiler industry.

**Abstract:**

Probiotics are a prominent alternative to antibiotics in antimicrobial-free broiler farming. To assess the effect of *Sphingomonas* sp. Z392 (isolated and identified) on broiler growth, 600 one-day-old Kebao broiler chickens were randomly divided into a control group and an experimental group. Each group had three replicates, with 100 broiler chickens being raised in each replicate. Regarding the experimental group of broiler chickens, 4.0 × 10^5^ CFU/mL of *Sphingomonas* Z392 was added to their drinking water. Then, the changes in broiler body weight, the EPI, intestinal histological structure, and gut microbiota were examined. The results show that the supplementation of the broilers’ drinking water with 4 × 10^5^ CFU/mL of *Sphingomonas* Z392 resulted in an increase in the relative abundance of *Lactobacillus*, *Bacteroides*, *Lachnospiraceae*, *Aminobacterium*, *Oribacterium*, *Christensenellaceae*, *Faecalibacterium*, *Barnesiella*, *Ruminococcus*, *Parabacteroides*, *Phascolarctobacterium*, *Butyricicoccaceae*, and *Caproiciproducens*, which have been reported to be positively correlated with the improved digestion and absorption of broiler chickens. The relative abundance of *Odoribacter*, *Alistipes*, *Parabacteroides*, and *Rikenellaceae* increased, and these have been reported to be negatively correlated with the occurrence of intestinal diseases. The relative abundance of *Campylobacter*, *Shigella Castellani*, *Bilophila*, *Campylobacter*, *Clostridia*, and *Anaerotruncus* decreased, and these have been reported to be positively correlated with the occurrence of intestinal diseases. At the same time, the following also increased: the integrity of small intestinal villus morphology; the number of goblet cells in small intestinal epithelial cells; the health of the mitochondria in the cytoplasm of jejunal villous epithelial cells; the number of lysosomes in the cytoplasm of goblet cells in the small intestinal epithelium, ileal villous epithelial cells, and mitochondria in the cytoplasm of large intestinal villous epithelial cells; the VH/CD of the ileum; and digestive, absorption, and defense capabilities. In particular, the final weight increased by 4.33%, and the EPI increased by 10.10%. Therefore, the supplementation of broiler drinking water with *Sphingomonas* generated better economic benefits from the broiler chickens.

## 1. Introduction

The use of feed additives has increased the success rate of broiler farming. Antibiotics, a feed additive, can selectively inhibit unwanted flora in the gut, thereby aiding in promoting growth [1,2]. However, the selection and application of feed additives have become more rigorous with the increasing attention to food safety. Green and healthy alternatives to antibiotics have been developed at this stage, such as probiotics, prebiotics, enzymes, and acidifiers [1]. Among them, probiotics are widely used as alternatives to antibiotics due to their unique advantages [3]. They can increase the number of beneficial microorganisms in the intestine, promote host intestinal health, and assist in the digestion and absorption of nutrients by producing hydrolytic enzymes [4]. Probiotics can also enhance immunity by regulating the gut microbiota and reducing the colonization of pathogenic bacteria in the intestine, thereby improving the structure of the gut microbiota [2,5]. In broiler farming, microorganisms such as *Lactobacillus*, *Lactobacillus bulgaricus*, *Lactobacillus acidophilus*, *Lactobacillus casei*, *Lactobacillus salivarius*, *Streptococcus thermophilus*, *Enterococcus faecalis*, *Lactobacillus plantarum*, *fecal coliform*, *Bifidobacterium*, *Aspergillus oryzae*, and *brewing yeast lactis* have been widely used [2,6,7,8,9,10]. Previous research has shown that probiotics such as *Lactobacillus casei* and *Streptococcus lactis* can increase the ratio of villus height to crypt depth in the small intestine [11,12]. These probiotics can help to improve intestinal health and feed utilization, as well as promote the growth and development of broilers [2,7].

*Sphingomonas* belongs to the Proteobacteria α-4 subclass, and it is a Gram-negative bacterium. The characteristics of the colony are as follows: yellow, round (diameter of 2–3 mm), neat edges, central protrusion, smooth and glossy surface, moist, and with a sticky texture. The cellular characteristics include short rod-shaped cells, a lack of spores, and unilateral polar flagella.

In our preliminary research, *Sphingomonas* Z392 was isolated from chicken cecum using a selective culture medium with ammonium nitrogen as the sole nitrogen source. It could use ammonium nitrogen in the intestine as a nitrogen source, convert it into nitrite nitrogen, and further convert nitrite nitrogen into nitrate nitrogen, thus, acting as a comammox bacterium [13].

To investigate the effect of *Sphingomonas* Z392 on the growth of broiler chickens, this study explored the role and mechanism of this strain in promoting the growth of broiler chickens through changes in body weight, the EPI, intestinal histological structure, and gut microbiota structure, providing a reference for the application of *Sphingomonas* Z392 in broiler production.

## 2. Materials and Methods

### 2.1. Experimental Design and Animal Breeding Management

In this study, 600 one-day-old Kebao broiler chickens, weighing 42 ± 2 g and provided by Henan Longhua Animal Husbandry Co., Ltd. (Zhumadian, China), were randomly divided into two groups: a control group (CK) and an experimental group (T). Each group had 3 replicates, with 100 broiler chickens raised in each replicate. The different broiler chicken replicates were kept in separate poultry houses sized 3.56 m × 2.38 m × 2.96 m (length × width × height). Regarding the experimental group of broiler chickens, 4.0 × 10^5^ CFU/mL of *Sphingomonas* Z392 was added to their drinking water. The feeding period for the broiler chickens was 42 days, and the feeding management was in accordance with the “Kebao Broiler Feeding Management Manual”, using indoor cage breeding with free feeding and drinking. Drinking water was added 6 times a day, with bacterial agents added according to the amount of water consumed each time, but the content of *Sphingomonas* Z392 was kept at 4.0 × 10^8^ CFU/L.

The use of experimental animals was approved by the Ethics Committee of Experimental Animals of Huanghuai University, and the provisions of the “Ethical Review of Experimental Animals of Huanghuai University” (Permit number: 202009220005) were strictly followed.

The feeding management of the Kebao broilers was divided into three stages: 1–18 days old, 19–33 days old, and 34–42 days old. The nutritional composition of the feed at different ages is shown in Table 1.

### 2.2. European Production Efficiency Factor for Each Group of Broilers

The sold age, survival rate, sold weight, and feed conversion ratio were calculated, and then the European production efficiency factor (EPI) for broiler farming was calculated: EPI = [survival rate × sold weight (kg)]/(feed conversion ratio × sold age) × 10,000 [14].

### 2.3. Observation Methods of Microstructure of Intestinal Tissue

At 42 d, the broiler chickens were executed by bleeding from the neck. The duodenum, jejunum, ileum, cecum, and rectum were collected in a timely manner, each with a length of 1–1.5 cm. After being fixed in formalin, they were embedded in paraffin, sliced, stained with hematoxylin–eosin (H&E), and sealed. Changes in the intestinal microstructure were observed with a microscope (Nikon Eclipse E100, Nikon, Tokyo, Japan) and photographed. Case Viewer 2.4 browsing software was used for analyses and observation. The integrity and lesions of tissues such as the epithelium, villi, Enterococcus, goblet cells, and submucosal glands were observed. In addition, the villus height and crypt depth of the small intestine were measured, and the ratio of the villus height to crypt depth was calculated. The detailed procedures were referenced from Naghi Shokri et al. [15] and Wang et al. [16].

### 2.4. Observation Method of Ultramicroscopic Structure of Intestinal Tissue

At 42 d, the broiler chickens were executed by bleeding from the neck. The duodenum, jejunum, ileum, cecum, and rectum were collected in a timely manner, each with a length of 0.5 cm. After being fixed with a fixative for electron microscopy, they were dehydrated at room temperature (15–25 °C), penetrated with and embedded in resin, polymerized, thinly sliced, stained, and observed using transmission electron microscopy (TEM) (HT7800, Hitachi Production Co., Ltd., Tokyo, Japan) for an image analysis. The focus was on observing the number of and pathological changes in organelles such as mitochondria, lysosomes, the endoplasmic reticulum, and ribosomes in intestinal epithelial cells. The detailed procedures were referenced from Zhao et al. [17].

### 2.5. Sequencing of Gut Microbiota

At 42 d, the broiler chickens were executed by bleeding from the neck. Five intestinal segments, the duodenum (du), jejunum (je), ileum (il), cecum (ce), and rectum (re), were collected in a timely and sterile manner, each with a length of 3 cm. After ligation at both ends, they were sent to Wuhan Servicebio Technology Co., Ltd. (Wuhan, China) for testing. Based on the Illumina platform, the genome of the gut microbiota was sequenced, and a bioinformatics analysis was performed using a two-end sequencing (PE250) method [18]. The gut microbiota of the broiler chickens were analyzed and determined using 16S rDNA sequencing technology.

The Alpha index was analyzed using QIIME2 2020.6.0 (https://qiime2.org/ (accessed on 20 December 2023)) analysis software; this software assesses information such as the richness and diversity of microbial communities in environmental samples through several diversity indices, employing commonly used metrics such as Chao, Shannon, and ACE, and it detects whether there is a significant difference in the index values between the control group and the treatment group. The beta diversity distance matrix was calculated using Qiime (2020.2.0) software, and the R (version 3.3.1) “vegan” package (version 2.4.3) was used for an NMDS analysis and mapping. The composition of the different subgroups at each phylum level was determined from the results of the taxonomic analysis: the dominant species contained in each sample at the phylum level and the relative abundance of each dominant species in the sample were determined.

### 2.6. Data Statistical Analysis

The experimental data were sorted out using Microsoft Excel 2013. The mean and standard deviation were calculated, and line and bar charts were drawn. A one-way analysis of variance was used to measure the differences between the experimental group and the control group.

## 3. Results

### 3.1. Comparison of EPI between Test and Control Groups of Broilers

During the experiment, the EPI of the broilers in each group was calculated, as shown in Table 2. Both groups of broiler chickens had European indices above 300, achieving profitability. Briefly, the addition of 4.0 × 10^5^ CFU/mL of *Sphingomonas* Z392 to drinking water significantly increased the final weight by 4.33% and the EPI by 10.10%, resulting in better economic benefits from the broilers.

### 3.2. Changes in Intestinal Microstructure

After the paraffin sectioning and H&E staining of various intestinal tissues, the results were observed under a binocular biological microscope, as shown in Appendix A. In the experimental group, the integrity of the small intestinal villi was enhanced, and the number of goblet cells in the epithelial cells of the small intestine increased. However, compared with the experimental group, the small intestinal villi of the broiler chickens in the control group became shorter and fragmented. The villi did not show their inherent morphology, especially those in the duodenum and jejunum, which were the most severely damaged. There were no visible differences in the folds and villi of the cecum and rectum in each group of broiler chickens.

### 3.3. Changes in the Ratio of Small Intestinal Villus Height to Crypt Depth

After measuring the villus height (VH) and crypt depth (CD) of the small intestine using Case Viewer 2.4 browsing software, the ratio of the villus height to crypt depth (VH/CD) of each segment of the small intestine was calculated, as shown in Figure 1. There was no significant difference in the VH/CD between the duodenum and jejunum groups (*p* > 0.05). The VH/CD ratio of the ileum in the experimental group (6.62 ± 0.06) was significantly higher than that in the control group (3.25 ± 0.55) (*p* < 0.05).

### 3.4. Changes in the Ultramicroscopic Structure of the Intestine

After the preparation and staining of ultra-thin sections of different intestinal segments, they were observed using transmission electron microscopy, as shown in Appendix A. In the test group, the addition of 4 × 10^5^ CFU/mL of *Sphingomonas* Z392 to the drinking water of the broiler chickens reduced the degree of mitochondrial swelling in the cytoplasm of jejunal villous epithelial cells; meanwhile, it increased the number of goblet cells in the small intestine epithelium, lysosomes in villous epithelial cells in the ileum, and mitochondria in villous epithelial cells in the large intestine.

### 3.5. 16S rRNA Gene Sequence Statistics

Illumina NovaSeq sequencing was performed, and the 16S rRNA gene sequence data of fecal samples from the duodenum, jejunum, ileum, cecum, and rectum of the two groups of chickens are presented in Table 3. Following quality control and the removal of chimeric sequences and those out of the target range, the average number of 16S rRNA gene sequences retained for analysis in each intestinal segment of the CK group was as follows: 62,608 in the duodenum, 69,890 in the jejunum, 66,047 in the ileum, 60,924 in the cecum, and 69,109 in the rectum. The average number of 16S rRNA gene sequences retained for analysis in each intestinal segment of the T group was as follows: 52,233 in the duodenum, 55,390 in the jejunum, 62,781 in the ileum, 67,302 in the cecum, and 53,054 in the rectum.

### 3.6. Alpha Diversity Analysis

As shown in Figure 2 and Table 4, the bioinformatics analysis of OTU found no significant difference in the Simpson index and Shannon index between the T and CK groups (*p* > 0.05). This finding suggests a high degree of consistency in the diversity of the gut microbiota between the CK and T groups.

### 3.7. Non-Metric Multidimensional Scale (NMDS) Analysis

A non-metric multidimensional scale (NMDS) analysis was performed on the bacterial communities at the genus level in five different parts, and the results are shown in Figure 3. There was a significant overlap between the CK and T groups in terms of the microbial communities in the duodenum, indicating that the differences in the microbial structure and composition in the duodenum of the CK and T groups are relatively small. However, the microbiota in the jejunum, ileum, cecum, and rectum of the CK and T groups did not overlap, indicating significant differences in the gut microbiota structure and composition in the jejunum, ileum, cecum, and rectum.

### 3.8. Analysis of Microbial Diversity at Phylum Level

The classification results of the gut microbiota in various intestinal segments from the different experimental groups are presented in Table 5 and Figure 4. At the phylum level, a total of 47 phyla were involved in the gut microbiota of the five parts. To facilitate the observation of relationships and species composition among the samples, a bar chart of the species distribution was constructed based on the abundance of each species within each sample. At the phylum level, there were 10 phyla with a higher relative abundance, namely, *Firmicutes*, *Bacteroidetes*, *Proteobacteria*, *Actinobacteria*, *Acidobacteriota*, *Chloroflexi*, *Cyanobacteria*, *Desulfobacteria*, *Gemmatimonadota*, and unclassified phyla.

### 3.9. Diversity Analysis of Microbial Communities at the Genus Level

To facilitate the observation of relationships between the samples and the composition of microbial species, a figure depicting the distribution of the microbial species for each sample was constructed based on the abundance of each microbial species within each sample (Figure 5). There were 10 genera with a higher relative abundance, namely, *Alistipes*, *Bacteroides*, *Christensenellaceae* R_7_group, *Faecalibacterium*, *Lactobacillus*, *Limosillactobacillus*, *Ruminococcus*_torques_group, unclassified *Lachnospiraceae*, unclassified *Oscillospiraceae*, and others, all of which demonstrate the intergroup differences.

#### 3.9.1. Differential Analysis of Duodenal Microbiota at the Genus Level

The content of bacterial strains in the duodenum and the changes in the different groups are shown in Table 6. Compared with the CK group, the relative abundance of *Lactobacillus*, *Bacteroides*, *Lachnospiraceae*, and *Aminobacterium* increased in the T group, and these are positively correlated with digestion and absorption. However, the relative abundance of *Odoribacter* increased, which is negatively correlated with the occurrence of intestinal diseases.

#### 3.9.2. Differential Analysis of Jejunum Microbiota at the Genus Level

The content of bacterial strains in the jejunum and the changes in the different groups are shown in Table 7. Compared with the CK group, the relative abundance of *Bacteroides*, *Lachnospiraceae*, *Oribacterium*, *Christensenellaceae*, *Proteiniphilum*, *Faecalibacterium*, *Barnesiella*, *Ruminococcus*, *Phascolarctobacterium*, and *Butyricicoccaceae* increased in the T group, and these are positively correlated with digestion and absorption. The relative abundance of *Alistipes* and *Paraacteroides* distasonis increased, and these are negatively correlated with the occurrence of intestinal diseases. The abundance of *Staphylococcus* in the jejunum also significantly increased.

#### 3.9.3. Differential Analysis of Ileum Microbiota at the Genus Level

The content of bacterial strains in the ileum and the changes in the different groups are shown in Table 8. Compared with the CK group, the T group exhibited an increase in the relative abundance of *Bacteroides*, *Aminobacterium*, and *Proteiniphilum*, which are positively correlated with digestion and absorption. The relative abundance of Parabacteroides increased, which is negatively correlated with the occurrence of intestinal diseases. The relative abundance of *Campylobacter*, *Escherichia*, and *Bilophila* decreased, and these are positively correlated with the occurrence of intestinal diseases.

#### 3.9.4. Differential Analysis of Cecal Microbiota at the Genus Level

The content of bacterial strains in the cecum and the changes in the different groups are shown in Table 9. Compared with the CK group, the T group exhibited an increase in the relative abundance of *Barnesiella*, *Oribacterium*, *Christensenellaceae*, and *Caproiciproducens*, which are positively correlated with digestion and absorption; the relative abundance of *Rikenellaceae* increased, which is negatively correlated with the occurrence of intestinal diseases; and the relative abundance of *Clostridia*, *Anaerotruncus*, and *Helicobacter* decreased, and these are positively correlated with the occurrence of intestinal diseases.

#### 3.9.5. Differential Analysis of Rectal Microbiota at the Genus Level

The content of bacterial strains in the rectum and the change folds in the different groups are shown in Table 10. Compared with the CK group, the T group exhibited an increase in the relative abundance of *Bacteroides*, *Oribacterium*, and *Ruminococcus*, which are positively correlated with digestion and absorption; the relative abundance of *Rikenellaceae* increased, which is negatively correlated with the occurrence of intestinal diseases; and the relative abundance of *Clostridia* decreased, which is positively correlated with the occurrence of intestinal diseases.

## 4. Discussion

### 4.1. Effects of Sphingomonas on the Histological Structure of the Intestine

The gastrointestinal tract (GIT) serves as the primary site for digestion and absorption, as well as for defense against harmful foreign microorganisms. Its function correlates positively with organelle numbers and health status [19]. The villi in the small intestine play a crucial role in nutrient absorption, and the height, integrity, and cell count of these villi significantly affect the digestive and absorption functions of the small intestine [20]. After adding 4 × 10^5^ CFU/mL of *Sphingomonas* Z392 to drinking water, broiler chickens exhibited an increased integrity of small intestinal villi; an augmented number of goblet cells in small intestinal epithelial cells; enhanced mitochondrial health in the cytoplasm of jejunal villous epithelial cells; and elevated counts of lysosomes in the cytoplasm of goblet cells in the small intestinal epithelium, ileal villous epithelial cells, and mitochondria in the cytoplasm of large intestinal villous epithelial cells. The VH/CD of the ileum also significantly increased (*p* < 0.05).

VH, CD, and VH/CD serve as three vital indicators for measuring the nutrient digestive and absorption abilities of the intestine, as well as its health status [19]. Intestinal villi can secrete mucus, protect and lubricate the mucosa, and promote the absorption of substances such as amino acids and glucose [21]. Thus, a higher VH indicates a stronger enzyme secretion capacity and better digestion and absorption abilities. CD reflects cell generation rates, with shallower crypts indicating increased cell maturation rates and enhanced intestinal secretion function [22,23]. A higher VH/CD value signifies stronger intestinal digestive and absorption capacities, while a decrease indicates weakened abilities [24,25]. The results of this experiment demonstrate that adding *Sphingomonas* to broiler chickens’ drinking water significantly increased the VH/CD ratio of the ileum and enhanced its digestive, absorption, and defense abilities. Moreover, the addition of *Sphingomonas* Z392 to drinking water led to a notable increase in intestinal probiotics such as *Lactobacillus*, which activate cell mitosis and induce intestinal epithelial cell proliferation by secreting short-chain fatty acids (SCFAs), resulting in increased intestinal villus length [26,27].

After adding mannan oligosaccharides, *Bacillus subtilis*, and *Bacillus licheniformis* to the diet, the height of the villi in the duodenum, jejunum, and ileum of broiler chickens increased, the number of goblet cells in the ileum segment increased, and the morphology of the gastrointestinal tract improved [28]. Our results are basically consistent with the results of this study, but the number of lysosomes in the cytoplasm of ileal villous epithelial cells and mitochondria in the cytoplasm of colorectal villous epithelial cells increased. The abovementioned research obtained better results than our study, possibly due to the combined effects of prebiotics and synbiotics on broiler chickens. This result also provides direction for our future research.

### 4.2. Effects of Sphingomonas on Gut Microbiota

Studying the relative abundance and diversity of the gut microbiota is crucial for understanding the impact of beneficial bacteria [28]. In our experiment, the Shannon curve’s horizontal width for multiple samples in each group was relatively large, indicating high species abundance in the samples. Additionally, the curve appeared relatively flat, suggesting a relatively uniform distribution of species across each group of samples. The coverage index for each group was 1, indicating that the different intestinal sequencing results in this experiment accurately reflect the true microbial composition of the samples.

The degree of intestinal health is determined by indicators such as microbial content and morphology, with the distribution of microorganisms varying among different intestinal segments [29]. The gut microbiota play a crucial role in maintaining animal health, immunity, and production performance, and probiotics have been shown to improve the microecological environment of broiler intestines [28]. Previous research has indicated that adding *Bacillus amyloliquefaciens* to the diet can increase the relative abundance of *Bacteroidetes*, *Butyricicoccaceae*, *Faecalibacterium*, *Heliobacillus*, *Lactobacillus*, *Parabacteroides*, and *Ruminococcus* in the gut of broiler chickens [30].

Our research found that adding 4 × 10^5^ CFU/mL *Sphingomonas* Z392 to the drinking water of broiler chickens increased the relative abundance of microorganisms, including *lactobacillus* in the duodenum; *Bacteroides* in the duodenum, jejunum, ileum, and rectum; *Lachnospiraceae* in the duodenum and jejunum; *Aminobacterium* in the duodenum and ileum; *Oribacterium* in the jejunum, cecum, and rectum; *Christensenellaceae* in the jejunum and cecum; *Proteiniphilum* in the jejunum and ileum; *Faecalibacterium* in the jejunum; *Barnesiella* in the jejunum and cecum; *Ruminococcus* in the jejunum and rectum; *Phascolarctobacterium* in the jejunum; *Butyricicoccaceae* in the jejunum; and *Caproiciproducens* in the cecum. The above microorganisms can promote the digestive and absorption abilities of broiler chickens. Furthermore, the microorganisms with reduced relative abundance include *Odoribacter* in the duodenum; *Alistipes* in the jejunum; *Parabacteroides* in the jejunum, ileum, and rectum; and *Rikenellaceae* in the cecum and rectum. These microorganisms can inhibit the occurrence of intestinal diseases [31,32].

The microorganisms *Lactobacillus*, *Lachnospiraceae*, *Oribacterium*, *Christensenellaceae*, *Faecalibacterium*, *Ruminococcus*, and *Caproiciproducens* belong to *Firmicutes*, which can hydrolyze starch and other sugars to produce butyrate, and which secrete short-chain fatty acids (SCFAs), such as acetate, propionate, butyrate, and lactate [33,34]. SCFAs play multiple roles in the intestine. They not only provide energy to the body but also reduce the types of harmful bacteria, stimulate the proliferation and differentiation of intestinal epithelial villous cells, increase the height of villi, expand the contact area between villi and chyme, and improve the digestive and absorption abilities of broilers [34]. In addition, the increase in beneficial microorganisms such as *Lactobacillus* in the intestine also helps to maintain the integrity of the intestinal structure and promote metabolism, thereby playing an important protective role as the first line of defense against pathogenic bacteria [35].

In addition, *Alistipes*, *Odoribacter*, *Rikenellaceae*, *Bacteroides*, and *Parabacteroides* belong to *Bacteroidetes*. They primarily digest grain feed and secrete mucin to protect and lubricate the intestines, exerting an anti-inflammatory effect [36,37,38]. *Bacteroidetes* can also enhance the disease resistance of broiler chickens by stimulating the immune system, increasing macrophage phagocytosis, and resisting the colonization of pathogenic bacteria [37]. The lithocholic acid secreted by *Odoribacter* promotes fat digestion and absorption, regulates blood lipids, promotes cell proliferation, exhibits anti-inflammatory properties, and protects the gastric mucosa. Additionally, the lithocholic acid produced by *Odoribacter* demonstrates excellent anti-inflammatory and antibacterial activities [39]. Even in small amounts, it effectively eliminates pathogenic microorganisms such as *Clostridium* and *Enterococcus* faecalis, reduces inflammation levels, and regulates the body’s immunity [39]. *Bacteroides*, a core member of the gut microbiota, regulates the host mucosal immune system, reduces inflammation, participates in carbon metabolism, and secretes SCFAs such as acetate and propionate [40]. *Rikenellaceae* in the intestine plays a certain role in protecting the body’s health and alleviating diseases [41].

*Aminobacterium Bacteroides*, a core member of the gut microbiota, regulates the host mucosal immune system, reduces inflammation, participates in carbon metabolism, and secretes short-chain fatty acids (SCFAs) such as acetate and propionate [42]. *Phascolarctobacterium* produces SCFAs, including acetate and propionate; reduces inflammation; and protects the intestinal mucosal barrier by decreasing the levels of lipopolysaccharide (LPS)-binding proteins and C-reactive protein (CRP) [43]. *Barnesiella* is associated with bile acid production and plays a crucial role in fat metabolism [44]. *Proteiniphilum* contributes to the conversion of acetic acid to butyric acid [45]. *Butyricicoccaceae* produces digestive enzymes in the intestine that break down starch, protein, and cellulose, promoting the digestion and absorption of these nutrients. Furthermore, butyric acid, the main metabolite of this genus, promotes the regeneration and repair of intestinal epithelial cells [46]. Therefore, *Butyricicoccaceae* significantly influences the intestinal microbiota’s structure, inhibits pathogenic bacteria, and promotes the growth of beneficial bacteria such as *Lactobacillus* [47].

Not only do probiotics enhance nutrient digestion and absorption and improve the breeding environment, but they also reduce pathogenic microorganisms and increase the relative abundance of anti-inflammatory microorganisms, thereby enhancing the body’s resistance to infections [30]. This study revealed that adding *Sphingomonas* Z392 to the drinking water of broiler chickens decreased the relative abundance of microorganisms such as *Campylobacter* in the jejunum and ileum, *Shigella Castellani* in the ileum, *Bilophila* in the ileum, *Clostridia* in the cecum and rectum, and *Anaerotruncus* in the cecum. These microorganisms are positively correlated with intestinal diseases [31,32].

However, after adding *Sphingomonas* to the drinking water of broiler chickens, the relative abundance of Staphylococcus increased in the jejunum, which can easily cause gastrointestinal diseases. However, there have been no reports in published studies indicating that adding probiotics to the diet can cause an increase in intestinal pathogenic bacteria. This may be due to the combined use of multiple probiotics, which inhibits the proliferation of pathogenic strains [2,48]. Therefore, when using *Sphingomonas*, it is necessary to use other microorganisms in combination to inhibit the proliferation of potential harmful bacteria, ensuring the health and breeding benefits of broilers.

### 4.3. Growth Promotion of Sphingomonas spp.

In this study, the relative abundance of microorganisms positively correlated with digestion and absorption and negatively correlated with intestinal diseases in broiler chickens was significantly increased by adding *Sphingomonas* spp. to drinking water during the feeding process. These microorganisms promote the digestion and absorption of nutrients in chyme by secreting digestive enzymes such as amylase, protease, glycosidase, and cellulase, or they produce SCFAs, which reduce the types of harmful bacteria and stimulate the proliferation and differentiation of intestinal epithelial villous cells. Both of these result in a high increase in villi, expanding the contact area between villi and chyme, and improving the digestion and absorption ability of broiler chickens [34]. Alternatively, secreted bile acids can promote fat digestion and absorption, regulate blood lipids, promote cell proliferation, exhibit anti-inflammatory properties, and protect the gastric mucosa. Meanwhile, the addition of these strains significantly reduced the degree of inflammation in the small intestine, ensuring its basic digestive and absorption functions, as well as promoting the digestion of nutrients such as sugars, proteins, fats, and inorganic salts [49]. Additionally, the number of goblet cells in the small intestine villous epithelium increased, playing an important role in maintaining intestinal health and promoting nutrient absorption. At the same time, the degree of mitochondrial swelling in the villous epithelial cells of the jejunum decreased, indicating a decrease in their degree of damage, thereby ensuring the digestive and absorption functions of the jejunum [49]. Importantly, the VH/CD ratio of the ileum significantly increased (*p* < 0.05), and the number of lysosomes in the villous epithelial cells increased. This significantly improved the digestive and decomposition abilities of the ileum, facilitating further absorption of nutrients in the ileum. Consistent with the results of previous studies, *Lactobacillus* and *Clostridium butyricum* also significantly affect the growth, development, and feed utilization of broiler chickens by regulating small intestinal VH and CD [2,50,51].

Additionally, due to the relatively short intestine of chickens, the digestion of chyme in the small intestine is not complete, so the cecum plays an important role in the chicken’s digestive system [52]. This study found that the number of mitochondria in cecal villous epithelial cells increased under the action of *Sphingomonas*, which promoted the absorption of nutrients by the cecum.

After adding 4.0 × 10^5^ CFU/mL of *Sphingomonas* Z392 to the drinking water of broiler chickens, the improvement in intestinal histology and microbial community structure not only enhanced the absorption function of their overall digestive tract but also improved their feed utilization rate and growth performance, resulting in an increase of 4.33% in the slaughter weight and 10.10% in the EPI, thus, achieving better economic benefits from the broiler chickens.

In practical applications, adding an appropriate amount of *Sphingomonas* Z392 to the drinking water of broilers can improve the histological structure and gut microbiota of the intestines, resulting in a 7.8% increase in the final body weight, a 0.02% increase in the survival rate, and a 0.05% decrease in the feed conversion rate, thereby bringing significant economic benefits. These results indicate that *Sphingomonas* Z392 has broad application prospects in broiler feeding, and it is expected to provide new strategies for the sustainable development of the global broiler industry.

## 5. Conclusions

The addition of 4 × 10^5^ CFU/mL of *Sphingomonas* sp. Z392 to drinking water improved the intestinal histology and flora structure, digestive and absorption functions, and feed utilization rate of broilers, and it resulted in a 4.33% increase in their final weight and a 10.10% increase in the EPI, which led to better economic benefits from the broilers. The use of *Sphingomonas* Z392 in the feeding process of broiler chickens is expected to provide new strategies for the sustainable development of the broiler industry.

## Figures and Tables

**Figure 1 animals-14-01920-f001:**
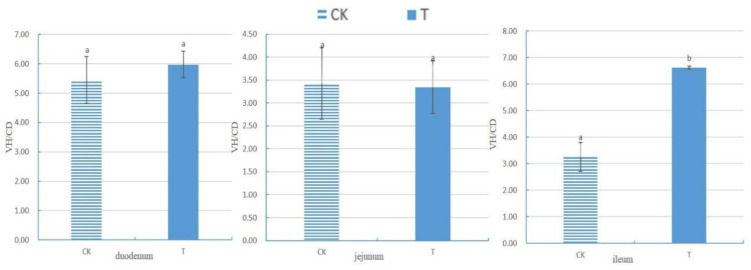
The ratio of small intestine villus height to crypt depth of chickens with (T) and without (CK) *Sphingomonas* Z392 in drinking water. Note: the same letter in the same column indicates no significant difference (*p* > 0.05) and the absence of identical letters indicates significant differences (*p* < 0.05).

**Figure 2 animals-14-01920-f002:**
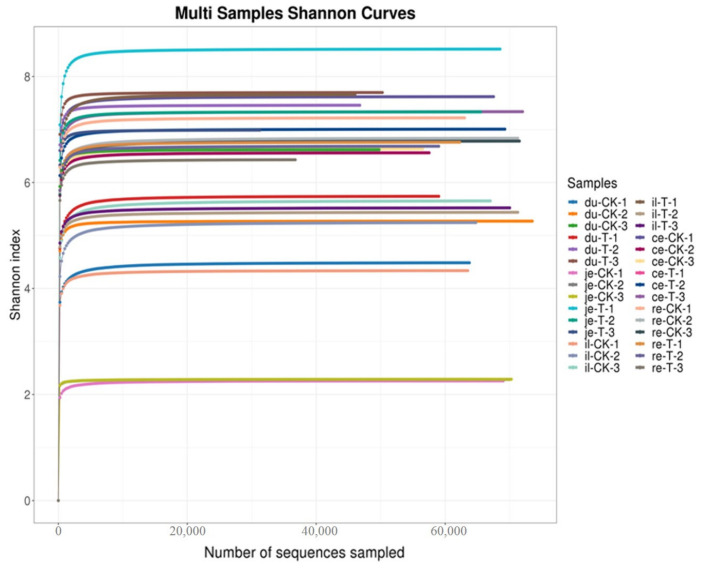
Multi-sample Shannon curves for broiler chickens with (T) and without (CK) *Sphingomonas* Z392 in drinking water. Note: the horizontal axis represents the number of sequencing samples randomly selected from a certain sample, and the vertical axis represents the Shannon index. As the sequencing quantity increases, more species are discovered, and, after species saturation, increasing the number of samples does not reveal new features.

**Figure 3 animals-14-01920-f003:**
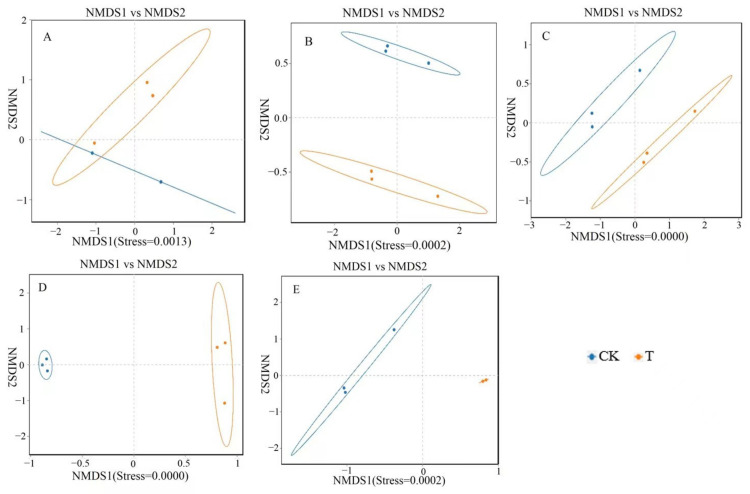
Non-metric multidimensional scale (NMDS) analysis of different intestinal segments from different groups of chickens with (T) and without (CK) *Sphingomonas* Z392 in drinking water. Note: 1. (**A**–**E**) show the results of the non-metric multidimensional scale analysis of the duodenum, empty field, ileum, cecum, and rectum, respectively. 2. The appearance of an obvious cross indicates that there is no significant difference in microbial structure and composition between groups, while the absence of a cross indicates significant differences in microbial structure and composition between groups. 3. Each point in the figure represents a sample; different colors represent different groups; the elliptical circle represents a 95% confidence ellipse (i.e., if there are 100 samples in the sample group, 95 will fall within it). When the stress is less than 0.1, it can be considered a good sorting and when the stress is less than 0.05, it has good representativeness. It is generally believed that, when the stress is less than 0.2, it indicates that the NMDS analysis has a certain level of reliability. The closer the samples on the coordinate map, the higher the similarity.

**Figure 4 animals-14-01920-f004:**
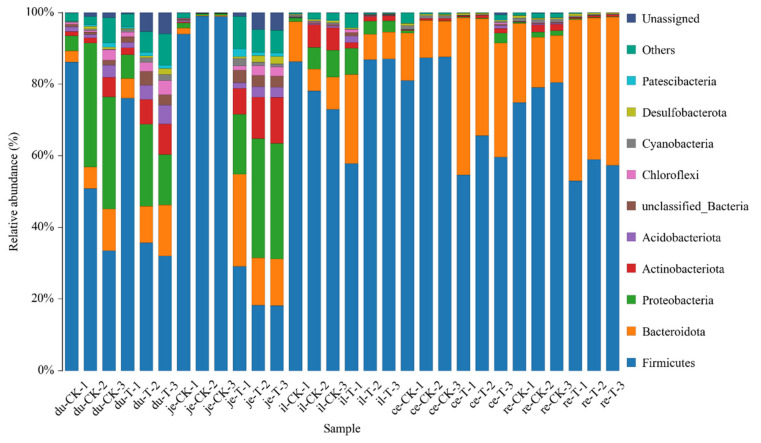
The content of microbial communities in different intestinal segments of chickens with (T) and without (CK) *Sphingomonas* Z392 in drinking water at the phylum level. Note: the horizontal axis represents the sample name and the vertical axis represents the relative abundance percentage (%). Different colors represent different species and stacked columns represent the top 10 taxonomic groups with relative abundance at each taxonomic level.

**Figure 5 animals-14-01920-f005:**
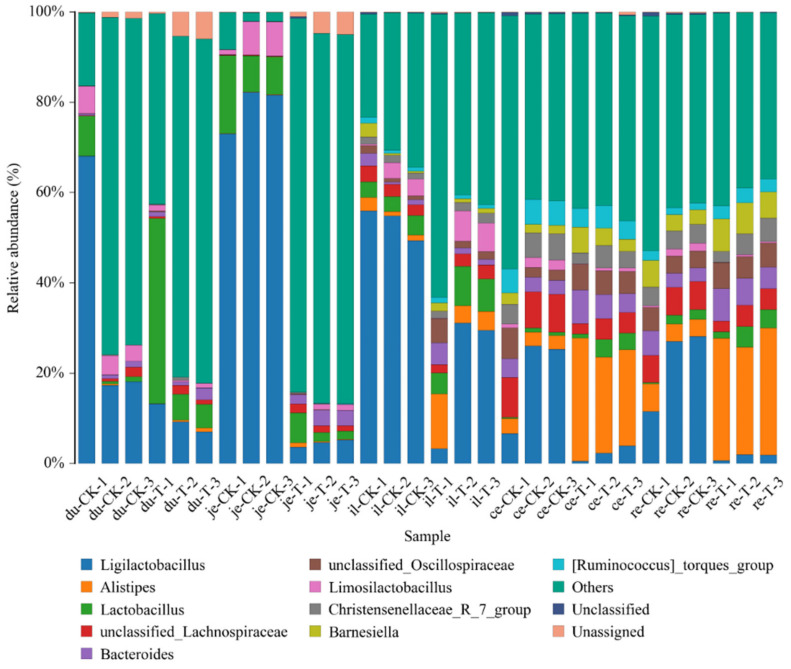
The content of different genera levels of microbial flora in different intestinal segments of chickens with (T) and without (CK) *Sphingomonas* Z392 in drinking water. Note: the horizontal axis represents the sample name and the vertical axis represents the relative abundance percentage (%). Different colors represent different species and stacked columns represent the top 10 taxonomic groups with relative abundance at each taxonomic level.

**Table 1 animals-14-01920-t001:** Nutrient composition of feed for broiler chickens at different ages.

Feeding Stage (Day)	Starter (1–18)	Grower (19–33)	Finisher (34–42)
Water (%)	12	12	12
Crude protein (%)	22	20	19
Crude fat (%)	4	5	7
Crude fiber (%)	3	3	3
Crude ash (%)	5	5	5
Calcium (%)	0.8	0.8	0.7
Total phosphorus (%)	0.65	0.55	0.6
Sodium chloride (%)	0.54	0.38	0.37
Methionine (%)	≤0.9	≤0.9	≤0.9
Metabolizable energy	12.5 MJ/kg	12.8 MJ/kg	13.2 MJ/kg

**Table 2 animals-14-01920-t002:** EPI of broiler chickens in each group.

	Sold Age/d	Sold Weight/kg	Survival Rate	Feed Conversion Ratio	EPI
CK	42	2.54 ± 0.006 ^a^	0.96 ± 0.006 ^a^	1.59 ± 0.012 ^a^	362.65 ± 3.47 ^a^
T	42	2.65 ± 0.01 ^b^	0.98 ± 0.006 ^b^	1.54 ± 0.006 ^b^	399.28 ± 1.63 ^b^

Note: The same letter in the same column indicates no significant difference (*p* > 0.05) and different letters indicate significant differences (*p* < 0.05).

**Table 3 animals-14-01920-t003:** Statistics of sample sequencing data processing results for broiler chickens with (T) and without (CK) *Sphingomonas* Z392 in drinking water.

	Intestinal Segment	Raw Reads	Clean Reads	Denoised Reads	Merged Reads	Non-Chimeric Reads
CK	du	69,584 ± 13,639.24	63,662 ± 12,142.49	63,583 ± 12,127.45	63,109 ± 12,058.75	62,608 ± 12,002.32
	je	79,087 ± 1484.33	73,337 ± 2068.34	73,278 ± 2057.47	73,062 ± 2111.19	69,890 ± 643.3
	il	79,965 ± 87.37	75,332 ± 205.46	74,958 ± 275.43	71,065 ± 740.99	66,047 ± 1952.38
	ce	79,953 ± 195.22	73,727 ± 2624.85	73,449 ± 2531.63	68,091 ± 1050.93	60,924 ± 5916.03
	re	80,060 ± 268.34	73,906 ± 2927.62	73,701 ± 2957.73	71,129 ± 3951.46	69,109 ± 4771.54
T	du	58,835 ± 7561.23	53,492 ± 6656.76	53,425 ± 6664.03	53,043 ± 6616.83	52,233 ± 6351.19
	je	64,566 ± 26,465.03	57,869 ± 22,754.19	57,758 ± 22,691.23	57,221 ± 22,541.65	55,390 ± 20,704.27
	il	71,205 ± 15,305.56	65,995 ± 15,788.91	65,858 ± 15,748.33	64,743 ± 15,233	62,781 ± 14,352.12
	ce	79,934 ± 65.73	74,194 ± 2152.66	73,969 ± 2236.57	70,843 ± 4343.02	67,302 ± 6587.09
	re	67,939 ± 21,015.84	62,847 ± 18,892.92	62,600 ± 18,770.72	58,192 ± 16,406.67	53,054 ± 14,027.97

Note: Raw reads, the number of raw reads obtained from sequencing; clean reads, the number of high-quality reads obtained after quality control of the original sequence; denoised reads, the number of clean reads after denoising; merged reads, the number of sequence entries obtained by concatenating denoised reads based on overlap; and non-chimeric reads, the final number of sequence entries after removing chimeras.

**Table 4 animals-14-01920-t004:** Alpha diversity analysis results for broiler chickens with (T) and without (CK) *Sphingomonas* Z392 in drinking water.

Sample	Feature	ACE	Chao1	Simpson	Shannon	PD_Whole_Tree	Coverage
CK	642 ± 346.47	645 ± 347.56	648 ± 349.01	1 ± 0.16	5 ± 1.85	79 ± 33.14	1
T	574 ± 183.03	576 ± 183.48	577 ± 183.79	1 ± 0.03	7 ± 0.85	93 ± 56.73	1

**Table 5 animals-14-01920-t005:** Classification and statistical results of microbial flora in different intestinal segments based on various classification levels of chickens with (T) and without (CK) *Sphingomonas* Z392 in drinking water.

Samples		Kingdom	Phylum	Class	Order	Family	Genus	Species
CK	du	2 ± 0	27 ± 2.52	50 ± 14.01	107 ± 35	177 ± 73.93	254 ± 126.89	276 ± 143.62
	je	2 ± 0	21 ± 3.21	35 ± 11.02	71 ± 25.51	109 ± 47.18	149 ± 71.77	160 ± 72.92
	il	2 ± 0.58	28 ± 6.08	55 ± 15.59	127 ± 49.39	226 ± 95.31	381 ± 177.79	417 ± 187.16
	ce	1 ± 0.58	13 ± 5.77	22 ± 10.97	45 ± 20.5	83 ± 38.73	149 ± 55.18	178 ± 55.07
	re	2 ± 0.58	25 ± 6.66	51 ± 20.21	115 ± 43.04	195 ± 75.92	306 ± 115.77	345 ± 125.3
T	du	2 ± 0	25 ± 3.21	50 ± 11.24	109 ± 32.08	170 ± 69.46	252 ± 129.69	269 ± 140.88
	je	2 ± 0	27 ± 3.79	51 ± 5.51	107 ± 13.05	169 ± 33.08	248 ± 67.1	264 ± 74.57
	il	2 ± 0	24 ± 2.08	50 ± 5.29	119 ± 10.79	200 ± 10.26	310 ± 22.5	348 ± 25.42
	ce	2 ± 0	17 ± 5	29 ± 11.14	63 ± 25.79	103 ± 30.12	175 ± 43.27	209 ± 44.6
	re	1 ± 0	12 ± 6.08	20 ± 10.97	46 ± 19.35	80 ± 27.73	143 ± 38.4	173 ± 41.88

Note: The numbers in the table represents the total number of reads covered by the sample at that level.

**Table 6 animals-14-01920-t006:** Change folds in the content of duodenal microbiota in chickens with (T) and without (CK) *Sphingomonas* Z392 in drinking water.

Genus	CK	T
*Aminobacterium*	9 ± 15.59	35 ± 60.04
*Bacteroides*	484 ± 224.89	755 ± 508.83
*Lactobacillus*	2195 ± 2958.18	9870 ± 12,485.8
*Lachnospiraceae*_NK4A136_group	537 ± 459.08	562 ± 335.41
*Odoribacter*	2 ± 3.21	102 ± 169.51
*Sphingomonas*	205 ± 80.18	597 ± 361.21

Note: The numbers in the table represent the total number of reads of the bacterial genus.

**Table 7 animals-14-01920-t007:** Analysis of change folds in jejunal microbiota in different groups with (T) and without (CK) *Sphingomonas* Z392 in drinking water.

Genus	CK	WT
*Alistipes*	56 ± 11.37	300 ± 343.23
*Bacteroides*	54 ± 20.66	1562 ± 625.47
*Barnesiella*	15 ± 13.45	20 ± 3.5
*Campylobacter*	53 ± 86.89	39 ± 9.07
*Christensenellaceae*_R_7_group	16 ± 1.15	82 ± 84.61
*Faecalibacterium*	32 ± 18.04	64 ± 54.6
*Lachnospiraceae*_NK4A136_grou	78 ± 24.76	914 ± 503.41
*Phascolarctobacterium*	0 ± 0	40 ± 69.28
*Proteiniphilum*	20 ± 8.33	232 ± 200
[*Ruminococcus*]_torques_group	25 ± 22.34	43 ± 41.55
*Staphylococcus*	2 ± 1.73	35 ± 28.21
*Sphingomonas*	35 ± 22.3	313 ± 128.91
unclassified_*Butyricicoccaceae*	0 ± 0	6 ± 2.05
unclassified_*Oscillospiraceae*	16 ± 1.15	82 ± 84.61

Note: The numbers in the table represent the total number of reads of the bacterial genus.

**Table 8 animals-14-01920-t008:** Change folds in ileal microbiota content in different groups of chickens with (T) and without (CK) *Sphingomonas* Z392 in drinking water.

Genus	CK	WT
*Aminobacterium*	2 ± 4.04	18 ± 31.18
*Bacteroides*	931 ± 729.93	1353 ± 775.55
*Bilophila*	58 ± 58.04	39 ± 13.58
*Campylobacter*	24 ± 33.86	29 ± 14.43
*Escherichia_Shigella*	63 ± 46.61	156 ± 122.19
*Parabacteroides*	572 ± 586.62	305 ± 20.53
*Proteiniphilum*	0 ± 0	75 ± 125.6
*Sphingomonadaceae*	42 ± 37.24	10 ± 16.74

Note: The numbers in the table represent the total number of reads of the bacterial genus.

**Table 9 animals-14-01920-t009:** Change folds in the content of cecal microbiota in different groups with (T) and without (CK) *Sphingomonas* Z392 in drinking water.

Genus	CK	WT
*Anaerotruncus*	25 ± 27.78	5 ± 5.03
*Barnesiella*	1281 ± 350.15	2652 ± 705.11
*Caproiciproducens*	0 ± 0	3 ± 5.77
*Christensenellaceae*_R_7_group	2143 ± 172.87	2522 ± 1006.54
*Rikenella*	0 ± 0	392 ± 82.25
*Sphingomonadaceae*	3 ± 5.2	0 ± 0
unclassified_*Clostridia*_UCG_014	2470 ± 663.49	1549 ± 293.3
unclassified_*Oscillospiraceae*	2387 ± 1956.92	3533 ± 107.68

Note: The numbers in the table represent the total number of reads of the bacterial genus.

**Table 10 animals-14-01920-t010:** Change folds of rectal microbiota content in different groups of chickens with (T) and without (CK) *Sphingomonas* Z392 in drinking water.

Genus	CK	WT
*Rikenella*	5 ± 4.16	377 ± 163.69
*Ruminococcus*	1126 ± 180.33	1589 ± 476.63
*Sphingomonadaceae*	3 ± 5.2	2 ± 4.04
unclassified_*Clostridia*_UCG_014	980 ± 303.77	305 ± 114.13
unclassified_*Oscillospiraceae*	112 ± 20.53	256 ± 103.01
uncultured_*Firmicutes_bacterium*	2609 ± 715.17	3247 ± 1414.46

Note: The numbers in the table represent the total number of reads of the bacterial genus.

## Data Availability

1. Experimental results of intestinal ultra microstructure. https://pan.baidu.com/s/1A_Zg1gtpj70T1ju0yHHhyg?pwd=dueg (accessed on 20 December 2023). Extract code: dueg. 2. Experimental results of 16S rRNA sequencing of gut microbiota. https://pan.baidu.com/s/16frep55lXGLevogLGqC2Dg?pwd=q0u3 (accessed on 15 March 2024). Extraction code: q0u.

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
