# Peer review of "Effects of Adding Sphingomonas Z392 to Drinking Water on Growth Performance, Intestinal Histological Structure, and Microbial Community of Broiler Chickens"

_animals, 2024, doi:10.3390/ani14131920_

Round 1
Reviewer 1 Report
Comments and Suggestions for Authors
Dear authors,
Please find below a few suggestions to improve your manuscript.
Introduction:
Paragraphs citing references 1-2... too general manner of speaking... these are well known aspects that do not require references.
paragraph with references 3-5... this is not an exhaustive list of feed additives for poultry. Please add all additives categories.
”Antibiotics, as additives, have been widely favored for their effectiveness in preventing diseases and promoting weight gain” [6]... antibiotics as feed additives were not used to prevent diseases, but to selectively inhibit the unwanted flora in the gut, so the only true statement is about their usage as growth promoters. Antibiotics are not prophylactic... immunological prophylaxis is carried on differently.
”China has now banned the use of antibiotics in feed additives”... it is good to specify the very moment since China have banned antibiotic usage as feed additives. In the EU, they are banned since 2005. How about in China. Also, a valid law regulation as reference would be useful to support the Chinese ban on these feed antibiotics.
Line 70 in Introduction... and improve meat quality.... this is not a direct related consequence... could be, but it is not the case of probiotics. It could be an indirect effect... If you do not change the very blunt statement about probiotics and meat quality, at least add a reference to support it.
In Introduction, there is not a direct reference to the dynamics or to the effects of probiotics on gut morphohistology and health-aspect of cell organelles, as you specified in the abstract and later in the results. Please include a paragraph to address the modification of the mucosae and epithelium under the effect of dietary probiotics in chicken. Thank you!
Matherial and methods:
Please use in Table 1 Diet instead of Feeding stage... Starter (1-18 days), Grower (19-33 days), Finisher (34-42 days). This is the regular division of diets in broilers. Also, please specify the Metabolisable energy content. It is also common in poultry and other monogastrics.
Section 2.2 - Only 5 broilers out of 300, from each group are to few and not representative for the weight gain dynamics. Please change with at least 30 individuals weekly (at least 10 individuals per replication per group). i am pretty sure you have weighed more than 5 per 300.
2.3 Sold age? Perhaps slaughter age... aren't they broilers farmed for slaughtering?
2.4. Euthanised? Perhaps slaughtered... aren't they broilers farmed for slaughtering in a slaughterhouse to capitalise their meat yield? Please add a reference about the histological technique that you have used to prepare the sealed smears, also please describe exactly what elements form the intestine you have studied (epithelium, villi, enterocytes, goblet cells, submucosal glandae etc. etc. etc.) and in which manner (depth, width, integrity, lesions etc. etc.). What equipment (optical microscope) have you used?
2.5. Please add reference and explain more accurately the way you have prepared samples form EM examination. Also, describe what elements of the cells (organelles etc.) you have examined and why...
2.6. Please add a reference for the sequencing method... also, it would be great to specify what kind of primers were used and who produced them.
2.7. SPSS 14.0 software was 139 used for analysis of variance. It is not necessary to specify both Excel and SPSS. SPSS also return descriptors you can calculate in Excel. Also, you did not specified the name of the statistical test you have used for the Analysis of variance. Please add info.
Please include in the Materials and Methods detailed information on>
* Alpha diversity methodology.
* NMDS Non Metric Multidimensional Scale Analysis methodology
* Analysis of Microbial Diversity at Phylum Level methodology
* Diversity Analysis of Microbial Communities at the Genus Level
The results are briefly and concise presented, all right.
Discussion> It is quite extreme to tell that villi edges secrete enzymes. These mostly come from the mucosal glands situated between the villi, deeper in in the cryptal mucosa that secrete intestinal juice. At the level of villi, the secretion is apocrine, made by the goblet cells and consist mainly in mucus with protective role in self-digestion.
Line 351> "Previous research indicates that adding probiotics to the diet can protect the gastro-intestinal tract from excessive feeding, increase VH and VH/CD in broiler chickens, and 352 reduce CD [20]." What means this prevention from excessive feeding??? Please elaborate or change the explanation/translation.
4.2. Effects of Sphingomonas on Gut Microbiota
From time to time you have used Actobacillus, please check everywhere.
4.3. It would be interesting to know if you studied a bit the caecal tonsil structure of the caeca... it would be expected that the lymphoid tissue is also stimulated by the new dietary probiotic hence better immunity and better growth rate. Also another lymphoid tissue in different gut segments should be stimulated by the probiotic addition.
Also, it is expected to emphasis a bit in the discussion area on the global effect of the provided probiotic on the growth performance (weight gain and especially FCR and survival rate). You could explain a bit the cause-effect cascade chains... How do you suppose the probiotic supplementation acts to eventually improve the broilers performance?
Conclusion> Please elaborate more, this is a conclusion of a short brief technical report. Please hypothesise on the consequences for long terms usage and on an eventual follow-up of the study...
Overall, congrats on your hard work, however, you have to elaborate more on the methodology and to the presentation of some results and on their discussion.
Thank you indeed!
Comments on the Quality of English LanguagePlease have your manuscript re-read by:
a) an expert in poultry science or poultry farming. There are some technical terms that are not translated or commanded quite well
b) an English native speaker.
Author Response
Thank you very much for reviewing our manuscript. We also greatly appreciate the reviewers for their complimentary comments and suggestions. We have revised the manuscript accordingly.

Reviewer 2 Report
Comments and Suggestions for Authors
Abstract
1. Please indicate a brief study design e.g number of treatments and replicate and the dosage used in water per liter
2. Delete the duplication of "this" in line 28
Introduction
line 60 - rather say alternatives not substitutes of AGP
line 69-70 Please add Ref
line 77-81 - Is it not supposed to be mentioned under material and methods?
Materials and methods
Any justifiable reasons why less than 5 replicates were used?
Please include the type of house was used and the pen size
line 96 add the dosage of the probiotic added to the water per L
Why was water given six times rather than at ad libitum?
line 108 - Why was only 5 chickens selected over 300 in a treatment. Was it not going to be better to select 5 per replicate?
Line 117 - rephrase "after 42 days" to at day 42
Line 117 - Please explain euthanized by bleeding"
Please also add the measurements (cm) of the intestinal collected
line 120 - Give details of the microscope used
line 124 - rephrase after 42 days
line 26 give range of room temperature
line 130 - rephrase after 42 days
line 118, 125, 131, Describe timely manner and sterile manner
Results
line 142 I suggest results to be presented in a table manner so we can see the significancy weights numbers
Discussion
line 332 - 333 - Add ref
line 351 - 353 - add one or two more ref
Please add similarities and contrast of your study to previous research and discuss those differences
ibitun
Comments on the Quality of English LanguageThe English is fine, it just needs minor errors to be removed
Author Response

(The authors gave the same response as above.)

Round 2
Reviewer 1 Report
Comments and Suggestions for Authors
Dear authors, thank you for improving your manuscript. You have clarified many aspects. From my point of view, it gained scientific value.
Congrats and now the final decision belong to the editors, as I agree with the publishing in the present form.
Best regards!
Comments on the Quality of English LanguageJust another re-reading for concision and to identify little misspels would be advisable. However, the quality of English improved substantially in comparison with the previous version.
Author Response
Thank you very much for reviewing our manuscript. We also greatly appreciate the reviewers for their complimentary comments and suggestions. We have revised the manuscript accordingly.
Introduction:
- Paragraphs citing references 1-2... too general manner of speaking... these are well known aspects that do not require references.
Answer: Thank you for your suggestion. This issue has been modified. “The use of feed additives has increased the success rate of broiler farming.”
And marked in red in the article.
- paragraph with references 3-5... this is not an exhaustive list of feed additives for poultry. Please add all additives categories.
Answer: Thank you for your suggestion. This issue has been modified. “Green and healthy alternatives to antibiotics had been developed at this stage, such as probiotics, prebiotics, enzymes and acidifiers. ”
And marked in red in the article.
- ”Antibiotics, as additives, have been widely favored for their effectiveness in preventing diseases and promoting weight gain” [6]... antibiotics as feed additives were not used to prevent diseases, but to selectively inhibit the unwanted flora in the gut, so the only true statement is about their usage as growth promoters. Antibiotics are not prophylactic... immunological prophylaxis is carried on differently.
Answer: Thank you for your suggestion. This issue has been modified. “Antibiotics as feed additives, could be selectively inhibit the unwanted flora in the gut, to achieve the effect of promoting growth. ”
And marked in red in the article.
- ”China has now banned the use of antibiotics in feed additives”... it is good to specify the very moment since China have banned antibiotic usage as feed additives. In the EU, they are banned since 2005. How about in China. Also, a valid law regulation as reference would be useful to support the Chinese ban on these feed antibiotics.
Answer: Thank you for your suggestion. The relevant content has been deleted.
- Line 70 in Introduction... and improve meat quality.... this is not a direct related consequence... could be, but it is not the case of probiotics. It could be an indirect effect... If you do not change the very blunt statement about probiotics and meat quality, at least add a reference to support it.
Answer: Thank you for your suggestion. The relevant content has been deleted.
- In Introduction, there is not a direct reference to the dynamics or to the effects of probiotics on gut morphohistology and health-aspect of cell organelles, as you specified in the abstract and later in the results. Please include a paragraph to address the modification of the mucosae and epithelium under the effect of dietary probiotics in chicken. Thank you!
Answer: Thank you for your suggestion. This issue has been modified. “Research has shown that probiotics can increase the ratio of villus height to crypt depth in the small intestine, such as Lactobacillus casei and Streptococcus lactis [11,12]. ”
And marked in red in the article.
Matherial and methods:
- Please use in Table 1 Diet instead of Feeding stage... Starter (1-18 days), Grower (19-33 days), Finisher (34-42 days). This is the regular division of diets in broilers. Also, please specify the Metabolisable energy content. It is also common in poultry and other monogastrics.
Answer: Thank you for your suggestion. Change Table 1 to: Nutrient composition of feed for broiler chickens at different ages. And the Metabolizable energy content has been added.
And marked in red in the article.
- Section 2.2 - Only 5 broilers out of 300, from each group are to few and not representative for the weight gain dynamics. Please change with at least 30 individuals weekly (at least 10 individuals per replication per group). i am pretty sure you have weighed more than 5 per 300.
Answer: Thank you for your suggestion. In order to reduce stress on animals and affect their normal growth. Therefore, during the sampling process, 5 samples were randomly selected for each repetition. However, we must pay attention to this issue in future experiments. Thank you indeed!
According to the suggestion of another reviewer, the method for representing weight every 7 days has been changed from the figure to the table.
Table 2. Weight changes of broiler chickens in each group.
1 |
7 |
14 |
21 |
28 |
35 |
42 |
|
CK |
42±0.71a |
185±4.64a |
460±3.67a |
820±6.16a |
1480±4.18a |
2049.8±3.03a |
2536.2±55.83a |
T |
42±0.71a |
200±5.07b |
480±3.96b |
880±0.24b |
1540±8.26b |
2095±120.71a |
2665±28.68b |
Note: The same letter on the right shoulder of the data in the same column indicates no significant difference (p>0.05); different letters indicate significant differences (p<0.05).
And marked in red in the article.
- 2.3 Sold age? Perhaps slaughter age... aren't they broilers farmed for slaughtering?
Answer: Thank you for your suggestion. In China, breeding enterprises and slaughtering enterprises are independent of each other. Breeding companies often mention the age of sale rather than the age of slaughter.
- 2.4. Euthanised? Perhaps slaughtered... aren't they broilers farmed for slaughtering in a slaughterhouse to capitalise their meat yield? Please add a reference about the histological technique that you have used to prepare the sealed smears, also please describe exactly what elements form the intestine you have studied (epithelium, villi, enterocytes, goblet cells, submucosal glandae etc. etc. etc.) and in which manner (depth, width, integrity, lesions etc. etc.). What equipment (optical microscope) have you used?
Answer: Thank you for your suggestion. For broiler chickens used for “Observing Intangible Microstructures and Ultra Microscopic Tissue” and “Sequencing of Gut Microbiota”, at 42 d, broiler chickens were executed by bleeding in the neck。 And other broilers were sold.
The microscope (Nikon Eclipse E100, Nikon Corporation of Japan)
Focus on observing the integrity and lesions of tissues such as Epithelium, Villi, Enterococces, Goblet cells, Submucosal Glandae, etc. In addition, the villus height and crypt depth of the small intestine were measured, and the ratio of villus height to crypt depth was calculated. The detailed procedures were referenced from Naghi Shokri et al. [15] and Wang et al. [16].
And marked in red in the article.
- 2.5. Please add reference and explain more accurately the way you have prepared samples form EM examination. Also, describe what elements of the cells (organelles etc.) you have examined and why...
Answer: Thank you for your suggestion. Focus on observing the number and pathological changes of organelles such as mitochondria, lysosomes, endoplasmic reticulum, and ribosomes in intestinal epithelial cells. The detailed procedures were referenced from Zhao et al. [17].
And marked in red in the article.
- 2.6. Please add a reference for the sequencing method... also, it would be great to specify what kind of primers were used and who produced them.
Answer: The reference for the sequencing method have been added (Trimmomatic: a flexible trimmer for Illumina sequence data.), the primers were produced by Wuhan Servicebio Technology Co., Ltd..
And marked in red in the article.
- 2.7. SPSS 14.0 software was 139 used for analysis of variance. It is not necessary to specify both Excel and SPSS. SPSS also return descriptors you can calculate in Excel. Also, you did not specified the name of the statistical test you have used for the Analysis of variance. Please add info.
Answer: Thank you for your suggestion. The experimental data were sorted out by Microsoft Excel 2013, and the mean and standard deviation were calculated, and draw line and bar charts. Use one-way analysis of variance to measure the differences between the experimental group and the control group.
And marked in red in the article.
- Please include in the Materials and Methods detailed information on
* Alpha diversity methodology.
Answer: We have added the methodology of Alpha diversity, and it has been marked by red words.
The Alpha index was analyzed using the QIIME2 (https://qiime2.org/) analysis software, which assesses information such as richness and diversity of microbial communities in environmental samples through several diversity indices, commonly used metrics such as chao, shannon, and ace, and detects whether there is a significant difference in the index values between the control group and the treatment group.
And marked in red in the article.
* NMDS Non Metric Multidimensional Scale Analysis methodology
Answer: We have added the methodology of Alpha diversity, and it has been marked by red words.
The beta diversity distance matrix was calculated using Qiime (2020.2.0) software, and the R language (version 3.3.1) vegan package (vsesion2.4.3) was used for NMDS analysis and mapping.
And marked in red in the article.
* Analysis of Microbial Diversity at Phylum Level methodology
Answer: We have added the methodology of Alpha diversity, and it has been marked by red words.
The composition of the different subgroups at each phylum level was obtained from the results of the taxonomic analysis: the dominant species contained in each sample at the phylum level; and the relative abundance of each dominant species in the sample.
And marked in red in the article.
* Diversity Analysis of Microbial Communities at the Genus Level
Answer: Diversity Analysis of Microbial Communities at the Genus Level is similar to the analysis of Microbial Diversity at Phylum Level methodology
And marked in red in the article.
- The results are briefly and concise presented, all right.
Answer: Thank you for your recognition.
Discussion
- It is quite extreme to tell that villi edges secrete enzymes. These mostly come from the mucosal glands situated between the villi, deeper in in the cryptal mucosa that secrete intestinal juice. At the level of villi, the secretion is apocrine, made by the goblet cells and consist mainly in mucus with protective role in self-digestion.
Answer: we deeply admire your profound knowledge, and we have benefited greatly. Intestinal villi can secrete mucus, protect and lubricate the mucosa, and promote the absorption of substances such as amino acids and glucose [14a].
- Line 351> "Previous research indicates that adding probiotics to the diet can protect the gastro-intestinal tract from excessive feeding, increase VH and VH/CD in broiler chickens, and 352 reduce CD [20]." What means this prevention from excessive feeding??? Please elaborate or change the explanation/translation.
Answer: Thank you for your suggestion. So sorry, it should be: Research has shown that adding probiotics to the die can reduce the stimulation of excess feeding on the gastrointestinal tract, increase VH and VH/CD in broiler chickens, and reduce CD.
But after careful consideration, it was not rigorous enough and has nothing to do with this study. Therefore, this section has been removed.
- 4.2. Effects of Sphingomonas on Gut Microbiota
From time to time you have used Actobacillus, please check everywhere.
Answer: Thank you for your suggestion. Actobacillus has been changed to Lactobacillus in the text.
And marked in red in the article.
- 4.3. It would be interesting to know if you studied a bit the caecal tonsil structure of the caeca... it would be expected that the lymphoid tissue is also stimulated by the new dietary probiotic hence better immunity and better growth rate. Also another lymphoid tissue in different gut segments should be stimulated by the probiotic addition.
Answer: Thank you for your suggestion. The impact of Sphingomonas on the disease resistance of broiler chickens is another research topic of the team, including the growth and decline of antibodies against major diseases, histological changes in thymus and cecal tonsils, and changes in immune indicators in serum.
- Also, it is expected to emphasis a bit in the discussion area on the global effect of the provided probiotic on the growth performance (weight gain and especially FCR and survival rate). You could explain a bit the cause-effect cascade chains... How do you suppose the probiotic supplementation acts to eventually improve the broilers performance?
Answer: Thank you for your suggestion.
In practical applications, adding an appropriate amount of Sphingomonas Z392 to the drinking water of broilers, could improve the histological structure and gut microbiota of the intestines, resulting in a 7.8% increase in final body weight, a 0.02% increase in survival rate, and a 0.05% decrease in feed conversion rate, thereby bringing significant economic benefits. These results indicate that Sphingomonas Z392 has broad application prospects in broiler feeding and is expected to provide new strategies for the sustainable development of the global broiler industry.
And marked in red in the article.
Conclusion
- Please elaborate more, this is a conclusion of a short brief technical report. Please hypothesise on the consequences for long terms usage and on an eventual follow-up of the study.
Answer: Thank you for your suggestion. The use of Sphingomonas Z392 in the feeding process of broiler chickens is expected to provide new strategies for the sustainable development of the broiler industry.
And marked in red in the article.
- Overall, congrats on your hard work, however, you have to elaborate more on the methodology and to the presentation of some results and on their discussion.
Answer: Thank you for your suggestion. All the questions you raised have been revised one by one. And marked in red in the article.
Thank you indeed!
Comments on the Quality of English Language
- Please have your manuscript re-read by:
- a) an expert in poultry science or poultry farming. There are some technical terms that are not translated or commanded quite well.
- b) an English native speaker.
Answer: Dr. Zhang Weina from the University of Western Australia and Dr. Chen Chong from Yangzhou University have been requested to revise the language and professional terminology throughout the entire text.

Reviewer 2 Report
Comments and Suggestions for Authors
Thank you for addressing the comments positively
Author Response
Thank you very much for reviewing our manuscript. We also greatly appreciate the reviewers for their complimentary comments and suggestions. We have revised the manuscript accordingly.
Abstract
- Please indicate a brief study design e.g number of treatments and replicate and the dosage used in water per liter
Answer: Thank you for your suggestion. The relevant questions have been added. And marked in red in the article.
- Delete the duplication of "this" in line 28.
Answer: Sorry, due to our negligence, we wrote an extra "this". Thank you for your guidance. The excess "this" has now been removed.
Introduction
3.line 60 - rather say alternatives not substitutes of AGP
Answer: Thank you for your suggestion. We have carefully studied the difference between "alternatives" and "substitutes" and realized their respective meanings. We have changed the word "substitutes" to "alternates" in the article.
4.line 69-70 Please add Ref
Answer: Thank you for your suggestion. The supporting references have been added.
5.line 77-81 - Is it not supposed to be mentioned under material and methods?
Answer: Thank you for your suggestion. After careful consideration, this part of the content is not related to this study. It has been removed now.
Materials and methods
6.Any justifiable reasons why less than 5 replicates were used?
Answer: Thank you for your suggestion. There are 100 chickens in each repetition, and these 100 chickens are 100 repetitions. Equivalent to 300 replicates for both the experimental and control groups. We sampled randomly during the experiment.
7.Please include the type of house was used and the pen size
Answer: Thank you for your suggestion. The different groups of experimental chickens were kept in individual cage in separated poultry house sized 3.56 m × 2.38 m × 2.96 m (length × width × height).
8.line 96 add the dosage of the probiotic added to the water per L
Answer: Thank you for your suggestion. The relevant questions have been added. And marked in red in the article.
- Why was water given six times rather than at ad libitum?
Answer: 1. Ensure the activity of the strain, 2. According to the regulations of feeding management, the laboratory should be inspected every 4 hours. During inspection, replace drinking water and add bacterial solution.
- line 108 - Why was only 5 chickens selected over 300 in a treatment. Was it not going to be better to select 5 per replicate?
Answer: Thank you for your suggestion. In order to reduce stress on animals and affect their normal growth. Therefore, during the sampling process, 5 samples were randomly selected for each repetition.
- Line 117 - rephrase "after 42 days" to at day 42
Answer: Thank you for your suggestion. This issue has been modified. And marked in red in the article.
- Line 117 - Please explain euthanized by bleeding"
Answer: Thank you for your suggestion. Broiler chickens were executed by bleeding in the neck. This issue has been modified. And marked in red in the article.
- Please also add the measurements (cm) of the intestinal collected
Answer: Observation of Intestinal Microstructure Tissue, each intestinal segment with a length of 1-1.5 cm. Observation of Intestinal Ultra Microscopic Structure Tissue, each intestinal segment with a length of 0.5 cm.
- line 120 - Give details of the microscope used
Answer: Thank you for your suggestion. This issue has been modified. And marked in red in the article.
- line 124 - rephrase after 42 days
Answer: Thank you for your suggestion. This issue has been modified. And marked in red in the article.
- line 26 give range of room temperature
Answer: Thank you for your suggestion. The range of room temperature is 15-25℃. And marked in red in the article.
- line 130 - rephrase after 42 days
Answer: Thank you for your suggestion. This issue has been modified. And marked in red in the article.
- line 118, 125, 131, Describe timely manner and sterile manner
Answer: Thank you for your suggestion.
"Timely Manner" means immediately starting dissection after confirming the death of the broiler.
"Sterile Manner" refers to removing feathers from the midline of the abdomen of broiler chicken, disinfecting it with 75% alcohol, making an incision with a disinfected surgical knife to expose the intestines, and then using disinfected surgical scissors and forceps to collect the intestines.
Results
- line 142 I suggest results to be presented in a table manner so we can see the significancy weights numbers
Answer: Thank you for your suggestion. The figure has been modified to a table,and all table and figure in the article have been reordered. And marked in red in the article.
Table 2. Weight changes of broiler chickens in each group.
1 |
7 |
14 |
21 |
28 |
35 |
42 |
|
CK |
42±0.71a |
185±4.64a |
460±3.67a |
820±6.16a |
1480±4.18a |
2049.8±3.03a |
2536.2±55.83a |
T |
42±0.71a |
200±5.07b |
480±3.96b |
880±10.24b |
1540±8.26b |
2095±120.71a |
2665±28.68b |
Discussion
- line 332 - 333 - Add ref
Answer: Thank you for your suggestion. We have carefully read the discussion section and added all the necessary references. And marked in red in the article.
- line 351 - 353 - add one or two more ref
Answer: Thank you for your suggestion. So sorry, it should be: Research has shown that adding probiotics to the die can reduce the stimulation of excess feeding on the gastrointestinal tract, increase VH and VH/CD in broiler chickens, and reduce CD.
But after careful consideration, it was not rigorous enough and has nothing to do with this study. Therefore, this section has been removed.
- Please add similarities and contrast of your study to previous research and discuss those differences
Answer: Thank you for your suggestion. Add 3 relevant discussions in the discussion.
After adding mannan oligosaccharides, Bacillus subtilis, and Bacillus licheniformis to the diet, the height of villi in the duodenum, jejunum, and ileum of broiler chickens increased, the number of goblet cells in the ileum segment increased, and the morphology of the gastrointestinal tract was improved [28]. The results are basically consistent with this study, but the number of lysosomes in the cytoplasm of ileal villous epithelial cells and mitochondria in the cytoplasm of colorectal villous epithelial cells increased. Their research showed better results than this study, possibly due to the combined effects of prebiotics and synbiotics on broiler chickens. This result also provides direction for our future research.
However, there have been no reports in published studies that adding probiotics to the diet can cause an increase in intestinal pathogenic bacteria. It may be due to the combined use of multiple probiotics, which inhibits the proliferation of pathogenic strains [2, 27a].
Consistent with the results of the study,
And marked in red in the article.
Comments on the Quality of English Language
- The English is fine, it just needs minor errors to be removed
Answer: Dr. Zhang Weina from the University of Western Australia and Dr. Chen Chong from Yangzhou University have been requested to revise the language and professional terminology throughout the entire text. And the paper (English-81965) has undergone English language editing by MDPI. The text has been checked for correct use of grammar and common technical terms, and edited to a level suitable for reporting research in a scholarly journal.
